# Mechanical Reliability Assessment of a Flexible Package Fabricated Using Laser-Assisted Bonding

**DOI:** 10.3390/mi14030601

**Published:** 2023-03-04

**Authors:** Xuan-Luc Le, Xuan-Bach Le, Yuhwan Hwangbo, Jiho Joo, Gwang-Mun Choi, Yong-Sung Eom, Kwang-Seong Choi, Sung-Hoon Choa

**Affiliations:** 1Graduate School of Nano IT Design Fusion, Seoul National University of Science and Technology, Seoul 01811, Republic of Korea; 2Faculty of Mechanical Engineering, Thuyloi University, 175 Tay Son, Dong Da, Hanoi 100000, Vietnam; 3Low-Carbon Integration Tech, Creative Research Section, ETRI, 218 Gajeong-ro, Yuseong-gu, Daejeon 34129, Republic of Korea

**Keywords:** flexible package, laser-assisted bonding, anisotropic solder paste, thermo-mechanical analysis, bending test

## Abstract

The aim of this study was to develop a flexible package technology using laser-assisted bonding (LAB) technology and an anisotropic solder paste (ASP) material ultimately to reduce the bonding temperature and enhance the flexibility and reliability of flexible devices. The heat transfer phenomena during the LAB process, mechanical deformation, and the flexibility of a flexible package were analyzed by experimental and numerical simulation methods. The flexible package was fabricated with a silicon chip and a polyimide (PI) substrate. When the laser beam was irradiated onto the flexible package, the temperatures of the solder increased very rapidly to 220 °C, high enough to melt the ASP solder, within 2.4 s. After the completion of irradiation, the temperature of the flexible package decreased quickly. It was found that the solder powder in ASP was completely melted and formed stable interconnections between the silicon chip and the copper pads, without thermal damage to the PI substrate. After the LAB process, the flexible package showed warpage of 80 μm, which was very small compared to the size of the flexible package. The stress of each component in the flexible package generated during the LAB process was also found to be very low. The flexible device was bent up to 7 mm without failure, and the flexibility can be improved further by reducing the thickness of the silicon chip. The bonding strength and environmental reliability tests also showed the excellent mechanical endurance of the flexible package.

## 1. Introduction

Flexible electronics have drawn much interest given their advantages and potential use in applications such as sensors, wearable devices, solar cells, displays, and batteries [1,2,3]. However, the rigidity and brittleness of silicon chips and other packaging components and materials represent a major hurdle that must be overcome before high-performance flexible devices can be realized [4,5]. Therefore, one of the main concerns to be addressed before flexible electronics can be realized is to develop a flexible packaging technology that determines the flexibility and reliability of the flexible devices. This type of flexible packaging also is the key factor in maintaining high performance capabilities, high throughput, and a low cost of flexible devices.

Currently, the packages for flexible electronics are developed using three main streams of technology: an ultra-thin silicon chip, a flexible substrate, and bonding technology that electrically connects the silicon chip and the substrate. There were various studies and remarkable achievements related to the fabrication of ultra-thin silicon chips, also known as ultra-thin chip (UTC) technology [6,7]. A flexible substrate is an essential and indispensable component to protect and support the thin silicon chip. Flexible substrates require several characteristics and properties, such as good flexibility, proper mechanical strength, lightness, a low cost, and thermal resistance. Various polymeric and metallic substrates were developed and used in flexible electronics to meet those stringent demands [8,9]. Among them, polyimide (PI) substrates are widely used due to their high mechanical robustness, flexibility, heat resistance, and low thermal deformation. Technology using a neutral plane concept is also widely used to improve the flexibility of flexible devices [10,11]. The neutral plane is the stress-free plane without tension or compression during bending.

A critical issue related to flexible device packaging is the bonding of the silicon chips to flexible polymer substrates with a low bonding temperature. Flip chip bonding technology is widely used in flexible electronics [12,13]. This technology uses solder materials to form an interconnection between the chip and the substrate using a reflow process. The melting temperature of common solder materials for bonding is, however, around 200 °C, which is relatively high compared to the glass transition temperatures of polymeric substrates. This will lead to the thermal deformation or the melting of these substrates [14]. In addition, the solidified solder bump formed after bonding is brittle and very fragile, making it vulnerable to bending stress. Therefore, various advanced bonding technologies and bonding materials were recently developed, involving the use of non-conductive paste (NCP) [15], anisotropic conductive film (ACF) [16,17], and anisotropic conductive paste (ACP) [18,19] materials. Recently, thermo-compression bonding [20], ultrasonic bonding [21], and laser bonding technologies [22] were also developed in place of the mass-reflow bonding method and are now widely used in the flexible materials industries to lower the bonding temperature and to realize uniform and stable bonding. 

Despite the different novel technologies developed and the quite remarkable progress in flexible electronics, there are still various technical issues for the practical applications of the flexible devices including the lower bonding temperature to minimize the damage of the flexible substrate and improving the environmental durability in high temperature and humidity. In our previous study [23], we briefly demonstrated the feasibility of a flexible device fabricated using a laser-assisted bonding (LAB) technology and anisotropic solder paste (ASP) material mainly focused on fabrication. 

In this study, we optimized the LAB fabrication conditions such as laser power and irradiation time and focused on the analysis of the mechanical reliability and flexibility of the flexible package. The LAB technology and the ASP bonding material were used to reduce thermal damage to the substrate and improve the reliability and flexibility of the flexible package. The thermo-mechanical deformation and stress of the flexible package after laser-assisted bonding were evaluated by experimental and numerical simulation methods. The bonding forces were evaluated. The flexibility of the fabricated package was also evaluated by bending tests and by a bending simulation. The environmental reliability tests were performed to validate the durability of the flexible package and bonding interface.

## 2. Experimental Procedure

### 2.1. Fabrication of the Flexible Package

To make the flexible device, a bare 8-inch silicon wafer was back-grinded using a wafer-grinding machine and polished to a thickness of 70 μm. The thin Si wafer was then cut to form a silicon chip 7 mm × 7 mm in size using a sawing machine. A copper laminated PI substrate 15 mm × 15 mm in size was used as the flexible substrate. A daisy chain pattern was fabricated on the silicon chip. The copper layer of the daisy chain pattern was coated onto the silicon chip using an electro-plating process. Figure 1 shows the daisy pattern on the PI substrate, and Figure 2 shows the fabrication process of the daisy pattern. To make the daisy chain pattern on the PI substrate, the copper layer was patterned and etched, after which the Ni/Au layer was electro-plated onto the copper layer. The thickness of the PI substrate and that of the copper layer were 70 μm and 25 μm, respectively. The thicknesses of the Ni and Au layer were 4 μm and 20 nm. The design and fabrication of the pad, mask, and silicon chip were described in detail in our previous study [23].

To bond the silicon chip and the PI substrate, an anisotropic solder paste (ASP) was screen-printed onto the metal electrode of the PI substrate using a screen printing machine. A stainless steel mask with a thickness of 50 μm was used during the screen printing process. The anisotropic solder paste is a mixture of solder powder, non-conductive polymer balls, and a thermosetting resin. The ASP contained Sn58Bi solder powder (5 vol.%) and non-conductive PMMA balls (6 vol.%) with a diameter of 20 μm. The thermosetting resin was composed of a base resin of epoxy, a curing agent, a reductant to remove oxide from the surface of the solder powder, and some additives. The ASP material in this study was developed and optimized for LAB process. No solvent or flux was present in the ASP material; thus, no vaporized gas was produced during the LAB process, and no cleaning process was necessary. During the bonding process, the electrical connection was achieved through the melted solder power, and the polymer PMMA balls acted as spacers.

After the screen printing process, the silicon chip and PI substrate were bonded using a laser-assisted bonding machine (Protec Inc., Korea, Anyang). A laser with a wavelength of 980 nm was used. The silicon chip and PI substrate were automatically aligned using an alignment system in the bonding machine. After the alignment step, a bonder header made of a transparent quartz plate was pressed at a pressure of 30 N (0.5 MPa). Figure 3 presents a schematic drawing of the laser-assisted bonding process with a homogenized rectangular laser beam. Laser light was transmitted through a quartz plate and irradiated onto the silicon chip. The silicon chip absorbed the laser energy from the laser beam and converted it into thermal energy. The heat in the silicon chip was conducted very rapidly to the copper electrodes and ASP material existing between the silicon chip and the PI substrate. The ASP material was instantaneously heated to its melting temperature, forming an electrical connection between the silicon chip and the PI substrate. After the laser irradiation step was finished, the flexible package was cooled to an ambient temperature, and the ASP material was solidified.

Before the LAB process, a series of experiments and numerical analyses were performed to optimize the LAB conditions. In particular, the optimization was focused on reducing the silicon chip temperature and bonding time as well as obtaining a temperature high enough to fully melt the solder. Through the optimization process, we finally applied a laser power of 160 W and laser irradiation time of 2 s. The size of the irradiated laser beam was equal to that of the substrate (225 mm^2^); therefore, the laser power density was 0.71 W/mm^2^. We assumed that the silicon chip was 70 mm thick and had an absorption coefficient of 29.4% [24]. The area of the silicon chip was 49 mm^2^. Therefore, the actual absorbed energy of the entire silicon die was as follows: 0.71 W/mm^2^ × 49 mm^2^ × 29.4% = 10.3 W.

### 2.2. Numerical Modeling

The laser-assisted bonding process of the silicon chip and PI substrate was analyzed using a finite element method (FEM). The heat transfer process and thermo-mechanical behavior of the flexible package during the laser bonding process were analyzed using ANSYS software. During the thermo-mechanical analysis, the deformation behavior of the flexible package and the mechanical stress of each component, which influenced the performance and reliability of the flexible package, were analyzed in detail.

Figure 4 depicts the structure and size of the numerical modeling used in this study, which were identical to those of the fabricated sample. The height and width of the printed ASP material were correspondingly 10 µm and 200 µm. Figure 5 illustrates the details of the temperature loading conditions in the simulation during the laser-assisted bonding process. First, the bottom stage below the flexible package was pre-heated to 80 °C. The laser was then irradiated onto the flexible package for 2.4 s. At the beginning of the laser irradiation step, the heat was transferred from the silicon die to the solder material very quickly. The solder then melted, and the electrical interconnection of the silicon chip and the substrate formed. Pressure force of 30 N was also applied to the flexible package during the laser irradiation step using a transparent quartz plate to ensure uniform and stable bonding during the overall bonding process, including the solder solidification and the cooling steps. As the laser irradiation time during the laser bonding process was very short, the heat transfer and bonding processes were analyzed by means of a transient process. The natural convection coefficient used was 9.75 W/m^2^ °C at 19 °C. 

Figure 6 shows the meshing structure of the FEM model used for the flexible package. The mesh structure was divided into square quadrilateral to increase the efficiency of the simulation. The FEM model had a total of 606,410 nodes and 214,736 elements. We assumed that the flexible package had no residual stress before the laser bonding process. After the heat transfer analysis, the results of the thermal analysis of the entire structure were transferred as input for the thermo-mechanical analysis. The boundary condition for the thermo-mechanical simulation was set as follows: one corner of the flexible package was limited, while the other three corners were allowed a free movement. All interfaces between the materials were considered to be perfect with good adhesion. The stress-free temperature condition was set to room temperature, i.e., 19 °C. The stage block below the flexible package was held constant at a temperature of 80 °C during the laser bonding process. The material properties used for simulation are shown in Table 1. The behavior of the materials was linear elastic. The numerical simulation of the solidification process of the solder material when cooling and after the irradiation step was conducted using a birth and death technology.

## 3. Results and Discussion

### 3.1. Heat Transfer Analysis

During the laser irradiation process, the temperature of the flexible device was measured using an infra-red (IR) camera and with a thin-film thermocouple (K type) sensor. Figure 7a shows the temperature profile measured with a IR camera. Figure 7b presents the changes in the temperature of the silicon chip during the laser bonding process as measured by the thermocouple. When laser irradiation began, the temperature of the silicon chip increased sharply to 250 °C for less than 0.5 s, and reached nearly 300 °C at 2.4 s. After laser irradiation, the temperature of the chip decreased quickly to the bonder stage temperature of 80 °C for less than 1 s. The temperature of the chip then decreased slowly via natural convection.

The changes in the temperature of the flexible package during the laser bonding process were also investigated via a FEM simulation. Figure 8a shows the simulated temperature distribution of the silicon chip and the substrate during the laser irradiation process. The highest temperature of the silicon chip was concentrated in the middle of the chip, and the temperature gradually decreased towards the outside areas. The maximum temperature of the silicon chip was 242 °C. The maximum temperature of the PI substrate was around 221 °C. Even though the possible damage of the active silicon devices during LAB process was not yet fully investigated, the maximum temperature of around 250 °C might have degraded or caused failure of the active silicon chips depending on the application. However, the heat exposure time was very short and localized, and thus, damage to the silicon chip was expected to be very minimal. That is why LAB technology recently started to be used in semiconductor packaging [25] and micro LED packaging [26]. 

Figure 8b exhibits the temperature changes over time in the silicon chip during the laser bonding process. The trend of the temperature change was very similar to that of the measurement results shown in Figure 7b. The maximum temperature of the silicon chip was maintained for 2.4 s during the irradiation process. After irradiation, the temperature of the silicon chip dropped rapidly to the stage temperature of 80 °C. This process was the cooling process after irradiation. It was evident that the bonding time of 3 s in LAB technology was much faster than that of the conventional reflow bonding process which usually requires for more than 10 min.

Figure 9 depicts the simulation results of the temperature distribution of the ASP solder area underneath the silicon chip during the laser irradiation process. The region shown in red in the figure is the ASP solder. The maximum temperature in the center region of the silicon chip, which is the red-color region, was around 242 °C. The highest temperature of the ASP solder was 220 °C. Considering that the melting temperature of the ASP solder is around 140 °C, the temperature during laser irradiation process was sufficient for solder to be fully melted and form the interconnection between the silicon chip and the substrate. These results indicate that heat was effectively transferred through the silicon chip to the solder in a very short time. Some of the heat was also transferred to the PI substrate. At the end of the irradiation step, the temperature dropped rapidly, causing the solder to solidify. It was found that the numerical simulation results were in good agreement with the experimental results. 

Figure 10 shows an image of the fabricated flexible device after laser bonding. The silicon chip was well bonded to the PI substrate, and the flexible device was relatively flat without any thermal deformation or damage to the PI substrate. We also conducted a cross-sectional analysis of the flexible package using an optical microscope. 

Figure 11 shows an image of the bonding area in which the silicon chip was bonded to the PI substrate. The Sn58Bi solder powder in ASP was completely melted and formed a contact area between the silicon chip and the copper pads. The squeezed polymer balls were also observed among the Sn58Bi solder areas. This result indicated that the bonding temperature during the laser bonding process was high enough to melt the solder power properly and that the electrical connection between the silicon chip and the PI substrate was properly formed. The bonding interface stability and bonding strength were evaluated using a shear tester (Nordson Dage-4000) with a speed of 0.13 mm/s. The total 10 samples were tested, and the shear strength was normalized with the cross-sectional bonding area. The average bonding strength was 21.3 MPa, showing a excellent bonding stability.

### 3.2. Warpage and Thermo-Mechanical Analysis

During the laser bonding process, each material with different coefficient of thermal expansions (CTEs) in the flexible package experienced uneven expansion and contraction. After the completion of the bonding step, thermo-mechanical residual stress was generated in the flexible package, causing the device to deform or warp. In this study, we investigated the thermo-mechanical behavior of the flexible package generated during laser bonding. Figure 12a shows the measurement results of the warpage in the diagonal direction of the flexible package at room temperature using a 3D digital image correlation technique (DIC) (Aramis 6M) after the completion of the bonding process. The warpage measured was 80 μm. Figure 12b shows the numerical simulation results of the warpage behavior in the diagonal direction of the flexible package. The simulated warpage value of the flexible package was approximately 90 μm. Considering the size of the flexible package (15 mm × 15 mm), the warpage value of the flexible package was minor.

Figure 13 shows the results of a comparison of the warpage behavior between the experimental results and the simulation results of the flexible package. It was found that the flexible package was deformed in a convex (∩) or crying shape after the bonding process. The warpage behavior in the simulation results was in good agreement with that of experimental results. However, there were slight differences in the warpage values. It was thought that these differences were caused by the material properties used in the simulation, which differed from those in an actual situation. Additionally, we assumed that the original PI substrate before bonding was flat in the simulation. However, we found that the PI substrate was slightly deformed before bonding, possibly due to the handling process. 

During the laser bonding process, the components most vulnerable to residual stress were the brittle silicon chip and the interconnection region. Figure 14 shows the simulation results in the form of a stress distribution map of the silicon chip. The simulated maximum stress of the silicon chip was around 41 MPa. The maximum strain subjected to the silicon chip was 0.019%. This strain value was very small compared to the fracture strain of the silicon material (0.8%) [27], meaning that there was no chance for cracks or damage to arise during the laser bonding process. Figure 15 shows the simulated von Mises stress distribution of the solder joint region generated during the laser bonding process. The maximum stress occurred at the top edge of the solder, which was near the interface between the solder and the silicon chip. The maximum stress of the solder was 1.8 MPa, which was very low. The yield strength of the solder was around 70 MPa; so, there was no plastic deformation or failure of the solder material during the laser process. The solder was, therefore, well formed and showed good bonding after the laser bonding process.

### 3.3. Bending Test and Simulation

In order to evaluate the flexibility of the package, bending tests of the flexible packages were conducted using a circular bar. Circular bars with different radii were used. Before the bending test, the electrical resistance of the contact pads of the daisy chain was measured using a four-point probe tester. The bending radius of the flexible package was changed from 10 to 6 mm. Four samples were tested in each test. After the bending test, the resistance of the flexible package was also measured in a flat state. Table 2 shows the results of the bending test. The electrical resistance value before the bending test was approximately 2.4 mΩ. Up to a bending radius of the 8 mm, the changes of the resistance were relatively small. As the bending radius decreased to 7 mm, the resistance increased to 3.4 mΩ. The reason for the increased resistance may have been due to micro-cracks generated in the silicon chip and bonding areas that were not generally visible under an optical microscope. Figure 16a shows an image of the bending test of the flexible package at a bending radius of 6 mm. For the bending radius of 6 mm, the resistance nearly doubled to 5 mΩ and a longitudinal crack was observed on the surface of the silicon chip as shown in Figure 16b, indicating that the flexibility limit of the developed flexible package was 6 mm. 

A numerical bending simulation was also conducted, and the stress and strain in each component of the flexible package were analyzed. Figure 17 shows a strain distribution map of the silicon chip at a bending radius of 6 mm. The maximum strain was located at the edge of the silicon chip, and the maximum strain value was 0.83%. As mentioned earlier, the fracture strain of the silicon material was around 0.8%. Therefore, the numerical simulation results clearly indicated that the bending strain of the silicon chip had nearly reached the fracture strain of the material at the bending radius of 6 mm, leading a fracture of the brittle silicon chip, in good agreement with the experimental result.

It was clear that the flexibility of the flexible package could be improved by reducing its thickness. For example, we intentionally reduced the thickness of the silicon chip from 70 μm to 30 μm, after which a numerical simulation was conducted. When the thickness of the silicon chip was 30 μm, the maximum strain generated when it was bent at 6 mm was 0.58%, which was much lower than the fracture strain. For the 30-μm-thick silicon chip, the flexible package could be bent at a bending radius of 4 mm, showing excellent flexibility.

Finally, to investigate the endurance of the flexible package and bonding material, the environmental reliability tests were performed for the flexible packages based on JEDEC standard. The reliability tests with high temperature and high humidity storage conditions (60 °C/90% RH) for 384 h and temperature cycling tests with −40 °C to 125 °C for 100 cycles were conducted. The changes of the electrical resistance of the contact pads were measured before and after the reliability tests. In each test, five samples were tested. It was found the changes in resistance of the samples after reliability tests were very small (less than 3%), indicating that the mechanical reliability of the developed flexible package was very good.

## 4. Conclusions

We developed a flexible packaging technology using laser-assisted bonding technology and an ASP bonding material to enhance the flexibility and reliability of a flexible device. A homogenized rectangular laser with a power of 160 W was used to irradiate the flexible package. Upon laser irradiation, the temperature of both the silicon chip and the solder material increased very quickly to 300 °C and 220 °C, respectively, at 2.4 s, which was high enough to melt the ASP solder. After irradiation, the temperature of the flexible package decreased quickly, and the solder was solidified. The results of a cross-sectional SEM analysis indicated that the solder powder in the ASP was completely melted to form a stable interconnection between the silicon chip and the copper pads, and there was no thermal damage of the PI substrate. The shear bonding strength was 21.3 MPa, which had excellent bonding interface strength. The warpage value of the flexible package was around 80 μm, which was very low compared to the size of the flexible package. The stress and strain of each component were also analyzed in a simulation. The stress subjected to the silicon chip and solder after the LAB process was very low, indicating that the potential for a failure or for plastic deformation was very low. Bending tests indicated that the flexible package could be bent to a bending radius of 7 mm without failure. The flexibility can be improved further if using a thinner silicon chip. The flexible package showed the good mechanical reliability for the high temperature and high humidity storage tests and thermal cycling tests.

## Figures and Tables

**Figure 1 micromachines-14-00601-f001:**
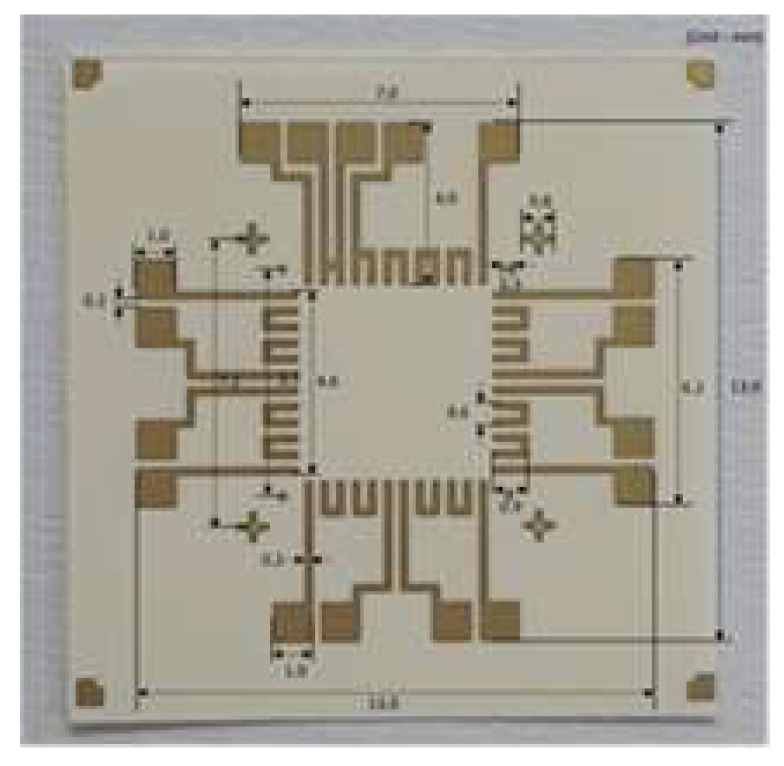
Daisy chain pattern on a polyimide substrate.

**Figure 2 micromachines-14-00601-f002:**
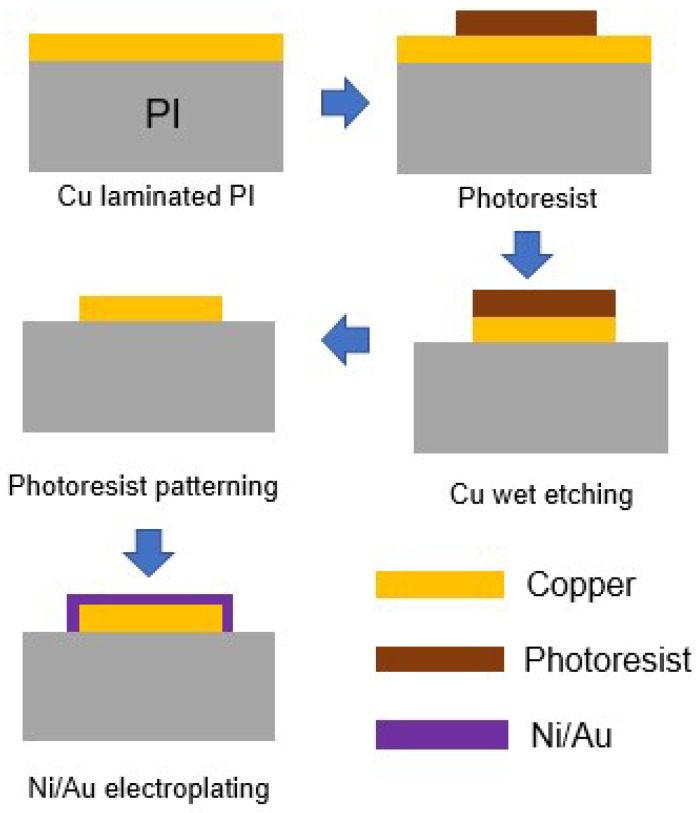
Fabrication process of the flexible substrate.

**Figure 3 micromachines-14-00601-f003:**
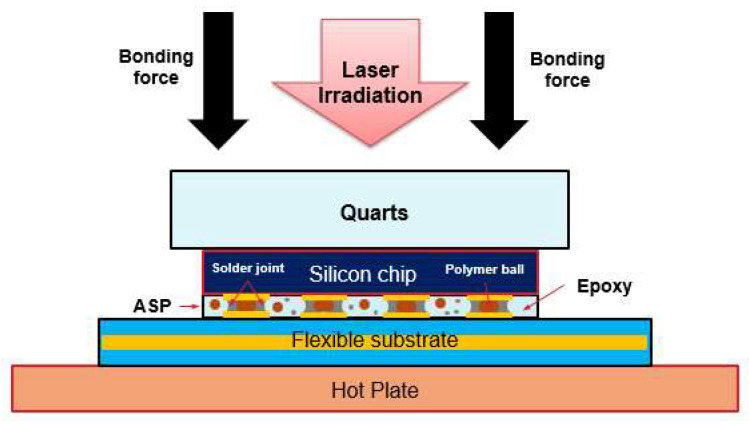
Schematic illustration of the laser bonding process of the flexible package.

**Figure 4 micromachines-14-00601-f004:**
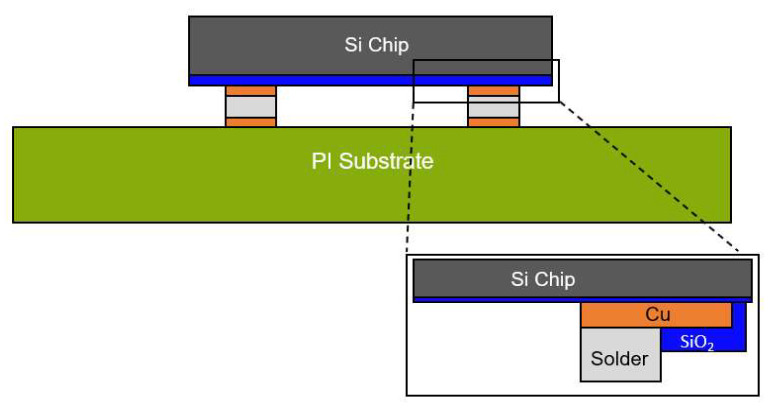
Structure of numerical modeling for the flexible package.

**Figure 5 micromachines-14-00601-f005:**
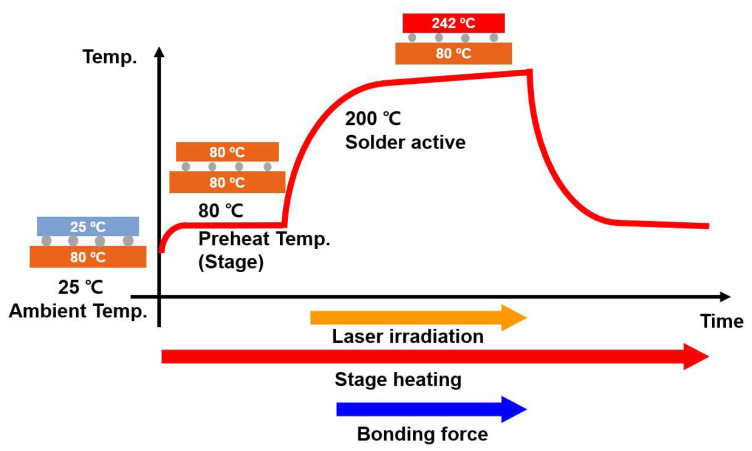
Temperature loading condition used in the numerical simulation.

**Figure 6 micromachines-14-00601-f006:**
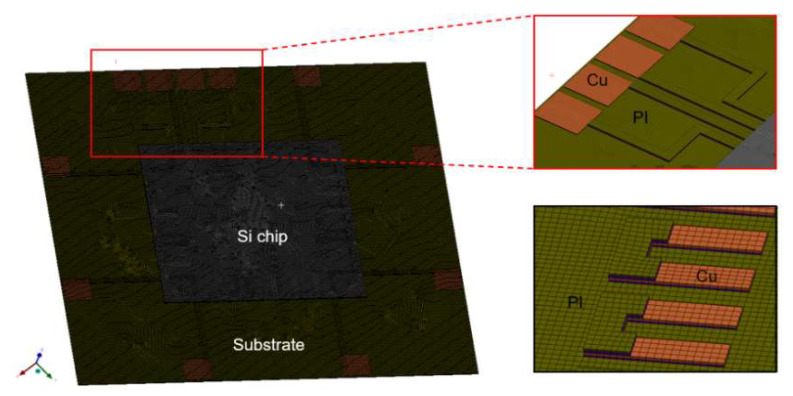
Meshing structure of the FEM model used for the flexible package.

**Figure 7 micromachines-14-00601-f007:**
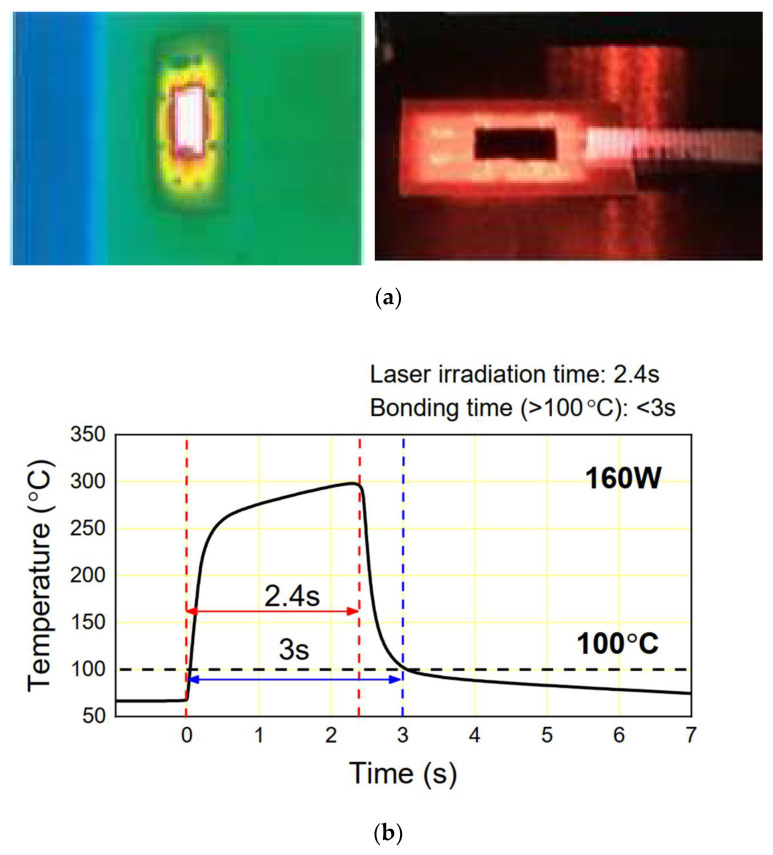
Temperature profile of the silicon chip during the laser-assisted bonding process: (**a**) temperature profile measured with the IR camera (**left**) and thermocouple (**right**). (**b**) temperature profile measured with the thermocouple.

**Figure 8 micromachines-14-00601-f008:**
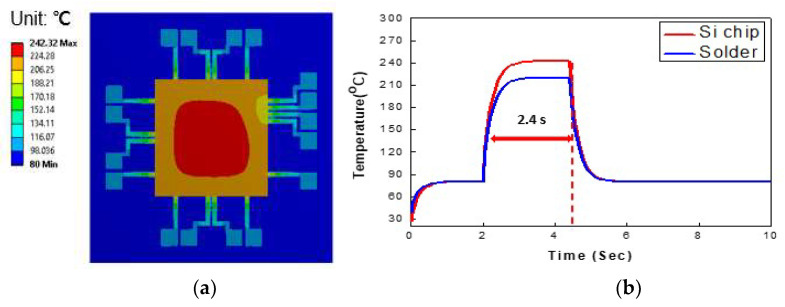
Simulated temperature distribution of the flexible package: (**a**) temperature distribution map of the silicon chip, and (**b**) temperature profile over time simulated with the FEA.

**Figure 9 micromachines-14-00601-f009:**
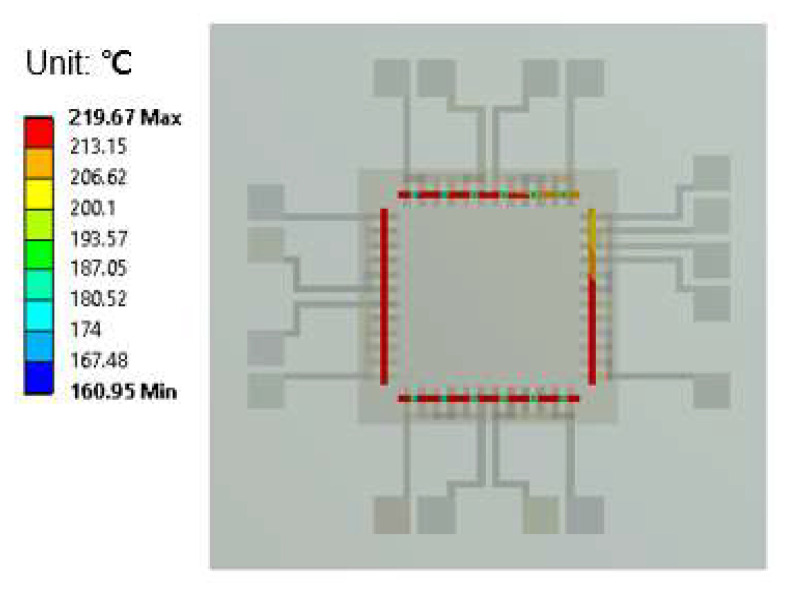
Simulation results of the temperature distribution of the PI substrate and the solder region.

**Figure 10 micromachines-14-00601-f010:**
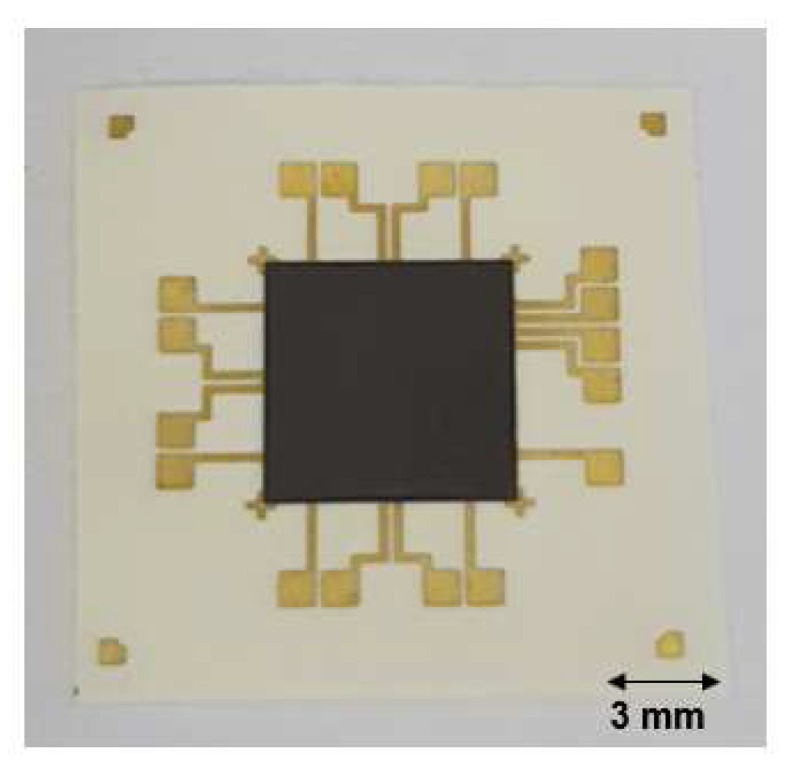
Image of the fabricated flexible device using the laser-assisted bonding technique.

**Figure 11 micromachines-14-00601-f011:**
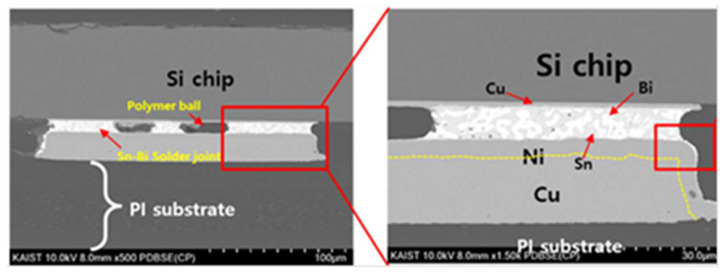
Cross-sectional image of the flexible package.

**Figure 12 micromachines-14-00601-f012:**
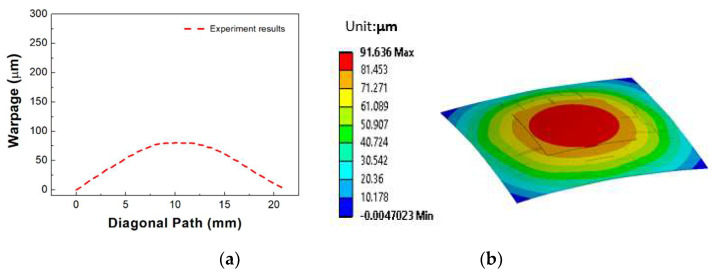
(**a**) Measurement results of the warpage of the flexible package, and (**b**) numerical simulation results of the warpage of the flexible package.

**Figure 13 micromachines-14-00601-f013:**
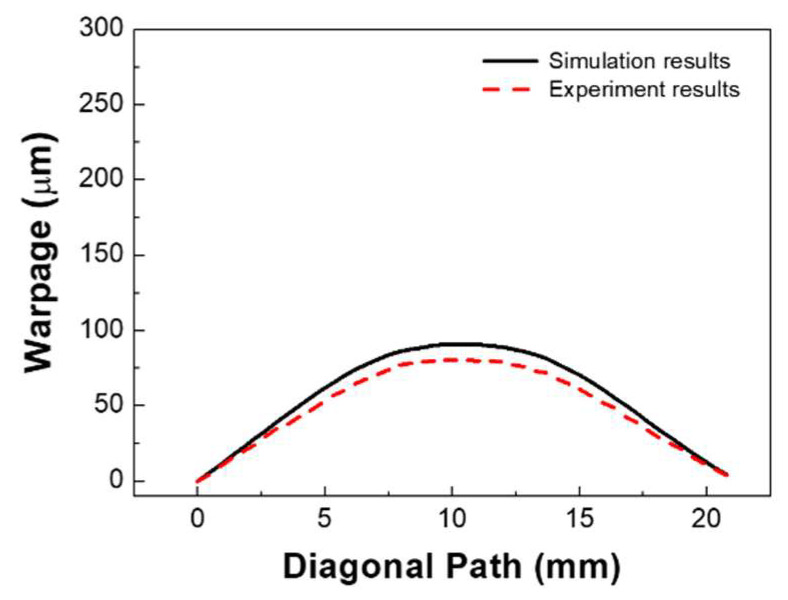
Comparison of the warpage of the flexible packaging between the experimental results and the simulation results.

**Figure 14 micromachines-14-00601-f014:**
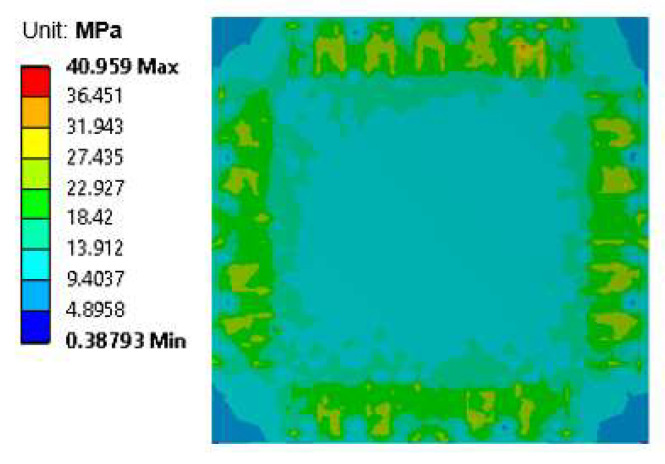
Stress distribution map of the silicon chip after the bonding process.

**Figure 15 micromachines-14-00601-f015:**
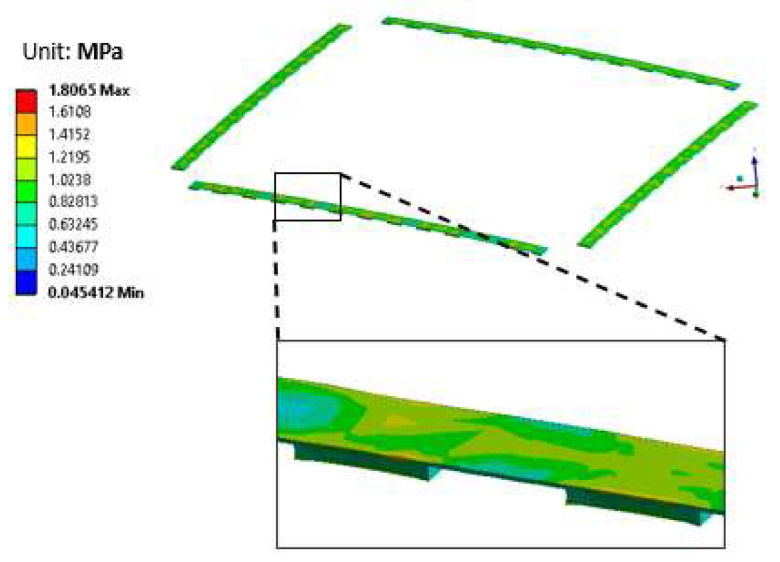
Von Mises stress distribution of the solder joint area.

**Figure 16 micromachines-14-00601-f016:**
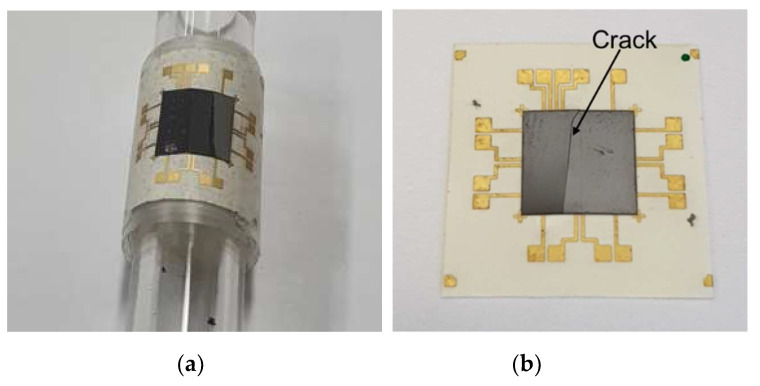
(**a**) Image of the bending test of the flexible device at a bending radius of 6 mm. (**b**) Optical image of crack in the silicon chip after bending test.

**Figure 17 micromachines-14-00601-f017:**
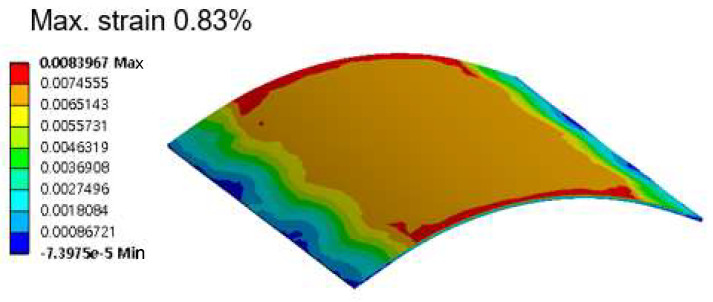
Maximum stress of the silicon chip at a bending radius of 6 mm.

**Table 1 micromachines-14-00601-t001:** Material properties used in the numerical simulation.

Component	Material	Density (kg/m^3^)	*E* (GPa)	ν	α (ppm/°C)	Thermal Conductivity (W/m × K)	Specific Heat (J/kg × K)
Die	Silicon	2300	131	0.28	2.8	124	794
Die mask	Silicon oxide	2170	66.3	0.15	0.55	1.3	680
Electrode	Copper	8960	128	0.34	16.5	398	390
Substrate	Adhesive	1150	0.69	0.4	50	0.17	1100
	PI	1420	4	0.35	20	0.12	1090
	Ni	8908	200	0.31	13.4	90.9	440
Solder	ASP solder	7360	1.36	0.4	62.5	1.045	167
	Epoxy	1150	2.5	0.3	50	0.17	1100

*E*: Young’s modulus; *ν*: Poisson’s ration; α: coefficient of thermal expansion.

**Table 2 micromachines-14-00601-t002:** Measurement of the electrical resistance for different bending radii.

Bending Radius (mm)	Electrical Resistance (mW)
Before bending (Flat state)	2.3
10	2.3
9	2.4
8	2.7
7	3.4
6	4.5

## Data Availability

The data are available as per request.

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
