# Peer review of "Mechanical Reliability Assessment of a Flexible Package Fabricated Using Laser-Assisted Bonding"

_micromachines, 2023, doi:10.3390/mi14030601_

Round 1
Reviewer 1 Report
The authors developed a laser-assisted bonding (LAB) process and ASP material to bond silicon thin chips to flexible (PI in this work) substrates for the applications of flexible devices. The overall process was successful where the laser irradiation quickly increased the temperature of the silicon chip and very fast heat transfer happened to melt the SAP. Together with the applied external force the bonding could be done easily. Characterizations and simulations indicated that the bonding strength, the mechanical properties, and the reliability of the bonded device are both satisfied. Overall, the manuscript was well structured and the quality meets the requirement of the journal. I have a few comments for the authors to consider which may further improve the quality of this work:
1. did the authors do the mechanical test to bend the bonded devices to a certain number of times until bending failure happened and then observe the changing of the bonding interface, the overall status of the device, and analyze the failure mechanism?
2. in section 3.3, it would be better if the reason of the increased resistance was analyzed and clearly expressed;
3. the temperature was quickly ramped up to more than 200 C while the melting point of the SAP is much lower. There might be an optimized temperature existed which is high enough to melt the SAP and at the same time as low as possible to minimize any harmful for the silicon chip which should have IC or electronic devices.
4. In Fig. 7(a) the resolutions of the images are very low. Higher quality is necessary.
Author Response
Dear Reviewer
I am very appreciated for your good and valuable comments.
I made a Answer file for reviewer's comment.
Please refer attached file.
Thank you very much.

Reviewer 2 Report
This paper researches the anisotropic solder paste (ASP) interconnection through laser-assisted bonding (LAB) for flexible packaging applications. The previous work [23] is extended in this manuscript with a detailed analysis and discussion of heat transfer, warpage, thermal-mechanical stress, the flexibility of the package, and longer time/more cycles reliability test in both experiment and simulation. The paper presents a comprehensive work of the literature review, design, process, simulation, characterization, and conclusion.
However, some points presented remain unclear, and some aspects could be described/explained in more detail. There are also some errors, which should be corrected.
For overall recommendation, if the manuscript is revised properly, it will be good to be published. Please check the attached PDF file for detailed comments, questions, and suggestions. Thank you.

Author Response
Dear Reviwer
We are very appreciated for your good and valuable comments.
We made a Answer file for reviewer's comment.
Please refer attached file.
Thank you very much

Round 2
Reviewer 2 Report
The authors addressed most of the comments and suggestions raised by the reviewers. Thank you. Three minor comments and suggestions:
- “The temperature profile in Fig.5 is quite different from that in Fig.7. Either Fig.5 should be revised or the difference should be discussed.” This comment is not addressed very well. I am asking why in Fig.5, which is the input of simulation, the laser irritation profile is quite different from Fig.7 and Fig.8.
- For the response to reviewer 1’s question 4. You claimed that the photo is an IR camera image. However, in Fig.7(a) of the manuscript, it is mentioned that that photo is the temperature profile of the thermocouple. Please double-check it.
- For future manuscripts, please mention the lines or location of the revisions in the authors’ response to reviewers so that it is easier for reviewers to find them in the revised manuscript. Also, the PDF file of the authors’ response is preferred.
Author Response
We are very appreciated for your good comments. We find some mistakes as you mentioned. We modified the manuscript and attached the Answer sheet.
